# Preparation of Biomass Carbon Composites MgO@ZnO@BC and Its Adsorption and Removal of Cu(II) and Pb(II) in Wastewater

**DOI:** 10.3390/molecules28196982

**Published:** 2023-10-08

**Authors:** Jie Yang, Qing Wei, Changan Tian, Dong Li, Hongming Li, Guangchao Qin, Kunhong Hu, Qinyan Zhang

**Affiliations:** 1School of Energy Materials and Chemical Engineering, Hefei University, Hefei 230601, China; 1187742334qq@163.com (Q.W.); li-hfxy@163.com (D.L.); 1610797071qq@163.com (H.L.); qgchao@163.com (G.Q.); kunhonghu@163.com (K.H.); 2CAS Key Laboratory of Crust-Mantle Materials and Environments, University of Science and Technology of China, Hefei 230026, China; 3School of Chemistry and Civil Engineering, Shaoguan University, Shaoguan 512005, China; 4Hefei Rantian Instrument Co., Ltd., Hefei 230031, China; muyuanouyang@163.com

**Keywords:** composite materials, metal oxide, adsorption, heavy metal, simulation

## Abstract

The ternary composite MgO@ZnO@BC was synthesized and characterized for the adsorption of Cu^2+^, Pb^2+^ heavy metal ions from wastewater. The results show that the addition of the MgO@ZnO@BC composite results in higher adsorption properties for Cu^2+^ and Pb^2+^, with a molar ratio of 5% 0.1 g, and maximum adsorption capacity (50.63 mg/g for Cu^2+^ and 61.46 mg/g for Pb^2+^). The Langmuir adsorption isotherm of the adsorption complex and the kinetics of adsorption are secondary kinetics. The adsorption of Cu^2+^ and Pb^2+^ was mainly chemisorption, accompanied by physical adsorption. This adsorption method fully conforms to the concepts of clean production and efficient waste utilization, providing a reference for the removal of heavy metal ions from wastewater and waste recycling using ternary composite materials.

## 1. Introduction

Over the years, large-scale industrialization has produced a large amount of heavy metals such as copper and lead wastewater discharged directly into water bodies causing water pollution [1]. Copper and lead ions in wastewater are toxic, carcinogenic, and not biodegradable [2]. According to the US Environmental Protection Agency (EPA) and the World Health Organization (WHO), the permissible levels of copper and lead ions in drinking water are 1.3 mg/L and 2 mg/L, 15 μg/L and 10 μg/L [3,4]. Pollution of water bodies has caused serious water shortages; therefore, the removal of copper and lead from wastewater is urgent [5]. There are many methods for treating heavy metals in wastewater. Technologies such as membrane filtration, ion exchange, liquid extraction, electro-dialysis, adsorption, and chemical precipitation have been widely used to remove heavy metal ions from water bodies [6,7,8,9]. Adsorption technology has become widely used because of its low price and high efficiency and stability without secondary pollution [10]; it has also become the most commonly used method for treating heavy metals in wastewater. According to the literature, several solvents such as Biomass carbon [11], metal oxides [12], zeolites [13], magnetite [14], chitosan [15], etc., have been used in the removal of heavy metal ions such as copper and lead from wastewater, and there are differences in their adsorption capacities. Among so many adsorbents, biomass-activated carbon has been widely studied because of its wide source, simple preparation, absence of secondary pollution, and low cost, which is in line with the current concept of green and clean production [16,17,18]. The poor absorption and recycling capacity of biomass charcoal limits its application in its heavy metal adsorption, so there is an urgent need to develop new technologies for the preparation of efficient adsorbents from biomass charcoal as raw material.

Banana is a tropical fruit that is eaten in large amounts, but its skin is discarded in large quantities without being fully used [19]. By making full use of these wastes to produce activated carbon, the problem of large amounts of banana peels that are not treated can be solved, and at the same time, the purpose of making full use of the waste can be achieved, in line with the concept of environmental protection [20]. In recent years, several researchers have started to modify banana peels into activated carbon for various applications such as lithium-sulfur batteries [21], removal of VOCs [22] and organic dyes [23]. In the treatment of heavy metals, because the poor removal effect hindered its application, some researchers modified it with acid and alkali, but the effect was not obvious, so new modifications are needed to improve adsorption performance [24,25,26], consistent with the sustainability rationale.

In recent years, metal oxides have gained the attention of researchers because of their green, low-cost, and excellent adsorption properties. This particular magnesium oxide is characterized by its abundant natural reserves, many active surface sites, environmental friendliness, and strong adsorption capacity [27,28]; therefore, it can become a promising and excellent adsorbent for treating heavy metal pollutants in wastewater. The hexagonal mesoporous nanosheets of MgO were prepared by a three-step method, and the maximum amount of nickel adsorption was 1684.25 mg/g [29]. Additionally, magnesium oxide has antibacterial activity [30]. This activity can lead to changes in adsorbent properties, instability of the synthesis process, and other properties that hinder application [31,32]. Therefore, the cost of the reagents used is high, the preparation steps are complicated, and dangerous byproducts are easily produced [33]. To avoid these problems, researchers have started to use biochar to immobilize magnesium oxide particle carriers, as well as increase adsorption activity and capacity [34,35].

For instance, ZnO is a semiconductor that is active in the ultraviolet region and has a large energy band [36]. Zinc oxide is widely used as an adsorbent because of its chemical stability, safety, environmental friendliness, as well as simple preparation [37,38]. Non-carbon structured ZnO adsorbents have received a lot of attention from researchers due to their excellent physical and mechanical properties [39,40]. By changing the morphology, introducing the lattice, and doping with metal oxides, the properties of ZnO can be improved [41,42,43,44]. ZnO nanoparticles have been prepared by the hydrothermal synthesis method and the maximum amount of chromium adsorption was 88.547 mg/g [45]. Additionally, zinc oxide is biocompatible, zinc oxide particles in pure water can be hazardous to health and therefore need to be fixed [46,47].

In summary, the main objective of this study was to test the feasibility of using MgO@ZnO@BC as an adsorbent to adsorb heavy metal ions from wastewater. The effect of ternary compounds on heavy metal ions (Cu^2+^ and Pb^2+^) in wastewater was explored for the first time. Therefore, in this work, ternary compounds of MgO@ZnO@BC were prepared by hydrothermal synthesis, characterized by SEM-EDX, XPS, FT-IR, and XRD. In addition, the morphology and adsorption mechanism of the materials were analyzed. The DQSE (Demand Question Strategy Extension) framework was proposed to provide an outlook on the prospects of application of ternary composites and to find a new direction for the development of adsorbent materials for efficient adsorption of heavy metals [48]. Recycling of magnesium ore and biomass waste is fully in line with the principles of environmental sustainability.

In Figure 1, we can see sources of heavy metal contaminants. Increased legal awareness will contribute to the impact of heavy metals on agroecology, food chain, and human health.

## 2. Results and Discussion

### 2.1. Characterization of MgO@ZnO@BC

The morphology and composition of the ternary composite adsorbent MgO@ZnO@BC were analyzed by SEM-EDX. In Figure 2, we can see that the morphology shows a large lamellar structure with an irregular distribution of magnesium oxide. The sample also shows a three-dimensional flowerlike structure, which is stacked and uniformly dispersed [49]. It can also be seen that the ternary MgO@ZnO@BC composite has a spherical shape and a highly porous structure, which is one of the most critical factors in chemical applications, especially in the field of adsorption. From the SEM, we initially observed the morphology and structure of the sample and determined the percentage of each element in the material. The sample was characterized using EDS. The energy spectrum allowed us a visual representation of the fact that the material consists of four elements, C, O, Zn and Mg, indicating that the ternary composite was successfully synthesized by the hydrothermal synthesis method. Additionally, the adsorbent material has clear pores and a rich pore structure. Doped MgO and ZnO particles are uniformly distributed in the BC pores with small particles and no phenomenon of agglomeration. This is mainly due to the special doping method, which prevents the accumulation of MgO and ZnO flocs and prevents the clogging of BC pores, which is very beneficial for the adsorption of Cu^2+^ and Pb^2+^.

X-ray photoelectron spectroscopy can be used to analyze the content and presence of elements on the surface of materials. The XPS spectra are shown in Figure 3. From Figure 3a, we can see that the Mg 1s peak appears at 1310 eV, while the Osher peak appears at 307 eV and 350 eV; in addition to Mg, there are peaks of C and O, plus a nuclear level peak of additional Zn [50]. It can be seen in Figure 3b that the main functional groups of the MgO@ZnO@BC adsorbent before adsorption were C=C (284.6 eV), C-C (285 eV), C-OH (286 eV) and O=C-O (289 eV). Combined with XRD analysis, we can see that it is consistent with the results of the discussion, which show that the MgO@ZnO@BC ternary composites are successfully compounded between the various components.

To understand the pyrolytic behavior and reactions occurring during the pyrolysis of MgO@ZnO@BC ternary composites, we used a thermogravimetric analyzer to perform tests to analyze the mass loss at each stage and to analyze the compositional changes to derive the optimal pyrolysis temperature. From Figure 4a, we can see that the mass loss of the MgO@ZnO@BC ternary composites can be divided into three stages, namely the water loss stage at temperatures below 110 °C, the loss stage at 300–400 °C, and the carbonization stage above 700 °C [51]. The mass loss of the adsorbent is concentrated mainly in the second stage, with three obvious peaks at 60 °C, 200 °C, and 370 °C. They are caused by moisture loss, organic matter loss, and other factors, respectively. Therefore, the adsorbent maintains high thermal stability when pyrolyzed at 700 °C.

The crystal structures of the MgO@ZnO@BC ternary compounds were determined by XRD. As can be seen in Figure 4b, the corresponding diffraction peaks appear at (111), (200), (201), and (302). The grain size of MgO-ZnO can be calculated using the Scheele formula. The molar ratio of the substances in the samples was found to be almost the same by elemental analysis. In summary, the appearance of high-intensity peaks confirms that the product is the result of high-Crystal Structure Formation. It can also be seen from the figure that the MgO@ZnO@BC material prepared by this method has better crystallinity and fewer impurity peaks. Furthermore, it was found that the doping of MgO and ZnO almost disappeared in the (100) plane and the intensity of the diffraction peak in the (002) plane tended to weaken, which means that the addition of MgO and ZnO reduced the crystallinity of BC towards amorphous carbon with a larger specific surface area and porosity [27]. The appearance of new substances may be one of the reasons for the higher adsorption capacity of MgO@ZnO@BC, which may be related to the important role of functional groups on MgO@ZnO@BC.

The functional groups and chemical structures of the MgO@ZnO@BC ternary compounds were analyzed by FT-IR. It can be seen from Figure 4c that the characteristic O-H stretching vibration peak at 3600 cm^−1^ is due to the formation of water molecules bound to the surface of the sample [52]. The following are the observed vibration peaks: 2363 cm^−1^ is the C=O vibration peak, 1596 cm^−1^ is the C=C double bond vibration peak, 1401 cm^−1^ is the -CO_3_^2−^ antisymmetric stretching vibration peak, 1130 cm^−1^ is the C-O stretching vibration peak; 610 cm^−1^ is the Mg-O vibration peak, and 470 cm^−1^ is the Zn-O vibration peak. The vibrational peak of Mg-O occurs at 610 cm^−1^ and the characteristic vibrational peak of Zn-OH does near 470 cm^−1^ [53]. In addition, it can be seen that the number of bright functional groups of MgO@ZnO@BC was significantly increased, indicating that the adsorbent material has an effective oxidation effect on the adsorbent. Therefore, it can be concluded that the adsorbent material has a good adsorption effect on other substances as well.

### 2.2. Adsorption Experiment

#### Influencing Factors of Adsorption Performance

The change in adsorption effect with time is shown in Figure 5a,b. The amount of Cu^2+^ and Pb^2+^ adsorption in MgO@ZnO@BC compounds increases with time. It can be seen that the adsorption increased significantly in 2 h and reached equilibrium adsorption. It may be because the adsorption sites and pores of the composites have been filled, and heavy metals are difficult to adsorb [54]. The initial adsorption rate of MgO@ZnO@BC composites is very fast because the ternary composites have a large specific surface area. It also helps significantly to study the adsorption mechanism and the optimal adsorption time, providing the conditions for the adsorption kinetics. It can be seen that the MgO@ZnO@BC compounds with a molar ratio of 5% have the best adsorption effect.

The adsorption performance of Cu^2+^ and Pb^2+^ MgO@ZnO@BC composites with different concentrations was evaluated by adding 0.1 g of the MgO@ZnO@BC composite to several conical flasks containing 100 mL of Cu^2+^ and Pb^2+^ solutions. Cu(II) and Pb(II) concentrations ranged from 5 mg·L^−1^ to 50 mg·L^−1^ and were adsorbed under static conditions. The performance data are shown in Figure 5c,d. The amounts of adsorption of Cu^2+^ and Pb^2+^ increased with increasing concentrations of Cu^2+^ and Pb^2+^. The removal of Cu(II) and Pb(II) by MgO@ZnO@BC was highest when the initial concentration was 5 mg·L^−1^, and the removal rate gradually decreased with increasing concentrations of Cu(II) and Pb(II) in the solution. In addition, the adsorption tended to equilibrate with increasing Cu(II) and Pb(II) concentrations, which was because the active sites of MgO@ZnO@BC exceeded the adsorbable Cu and Pb amounts (II) at low concentrations of Cu(II) and Pb(II) wastewater, resulting in lower adsorption efficiency. When the concentration of Cu(II) and Pb(II) increased, the active sites for adsorption were insufficient, leading to an earlier adsorption equilibrium [55].

The effect of the addition of the adsorbent on the adsorption effect is shown in Figure 5e,f. Adsorption decreased with the increase in the amount of MgO@ZnO@BC addition, whereas the removal rate increased with the increase in the amount of MgO@ZnO@BC addition. The reason for this phenomenon is that the increase in the amount of the MgO@ZnO@BC adsorbent increases the effective adsorption surface area, thus increasing the adsorption of heavy metal ions [56]. According to the results of Li et al. reports, the adsorbent dosage at the intersection of adsorption and removal rate for each adsorbent material is the optimal adsorbent dosage [57]. The optimal amount of adsorption of Cu^2+^ and Pb^2+^ was 0.1 g/L, as shown in Figure 5e,f. To save material costs, the optimal dose of MgO@ZnO@BC in practical engineering applications is 0.1 g/L.

The removal rate was used as an index to examine the amount of adsorbent. Combined with the effect of Figure 5e,f on the removal rate, the results showed that MgO@ZnO@BC, with the additional amount of 0.1 g/L, had a better removal rate of Cu^2+^ and Pb^2+^, so it was chosen as the addition amount of MgO@ZnO@BC in the experiment.

The effect of temperature on the adsorption capacity is shown in Figure 5g. The adsorption of both Cu^2+^ and Pb^2+^ was slightly enhanced when the temperature increased from 20 °C to 30 °C. However, with a further increase in temperature, the adsorption tends to the equilibrium state. Under normal conditions, metal–ion adsorption is heat-absorbing, and the increase in temperature promotes adsorption. The ability of MgO@ZnO@BC to release a large amount of H^+^ at high temperatures is due to the temperature-responsive property of MgO@ZnO@BC at high temperatures. This structural transition is caused by the thermal deprotonation of the carboxyl group in MgO@ZnO@BC. By calculating the changes in the thermodynamic parameters ΔG and ΔH, it can be found that the adsorption is thermally absorbing and spontaneous below 40 °C [58]. The adsorption process from 40 °C to 60 °C is exothermic and non-spontaneous.

### 2.3. Thermodynamic Model

The adsorption isotherms of Cu^2+^ and Pb^2+^ ions in MgO@ZnO@BC are shown in Figure 6. The relevant values are shown in Table 1. As can be seen in Figure 6, the linear relationship between Ce/q_e_ and C_e_ is better, indicating that the Langmuir isotherm model can better explain the adsorption of Cu^2+^ and Pb^2+^ ions. The Langmuir isotherm model has the highest linear correlation coefficient (R_2_ = 0.99465 and 0.99329). To further confirm that the Langmuir adsorption model is the Cu^2+^ and Pb^2+^ adsorption on MgO@ZnO@BC, the removal of Cu^2+^ and Pb^2+^ was assumed to be in a homogeneous monomolecular layer with equal active sites and adsorption energies and no interaction between the adsorbates [50]. The maximum theoretical removal rates (QM) calculated by the Langmuir isotherm model were 50.63 mg/g for the maximum adsorption of Cu^2+^, 61.46 mg/g for the maximum adsorption of Pb^2+^ with a maximum adsorption of 61.46 mg/g. The maximum adsorption capacity and the actual adsorption capacity are multiply related, indicating that the process is unimolecular layer adsorption and the adsorption surface is not completely covered [59]. This indicates that the removal effect is significant in a well-performing material.

#### Adsorption Kinetics

The general results of the adsorption effect, the primary kinetics, and the secondary kinetics model fitting results for the adsorption process of Cu(II) and Pb(II) by MgO@ZnO@BC are shown in Figure 7a–f. It can be seen from the figures that the equilibrium was reached at the contact time of 25 h ((Cu^2+^) and (Pb^2+^)), and the metal removal rate did not change much with an additional further increase in time. This may be because the surface of the adsorbent was filled after reaching the equilibrium of adsorption and heavy metal ions were hardly adsorbed by the pores of the adsorbent. The results of fitting the parameters of the proposed primary kinetic models and the proposed secondary kinetic models were obtained. It can be seen that the correlation coefficient R_2_ obtained by fitting the equations of the proposed secondary kinetic model is larger than that obtained by fitting the equations of the proposed primary kinetics, which is close to one; the amount of adsorption q(e, cal) calculated from the equations of the proposed secondary kinetics agrees well with the experimental value q(e, cal) [60]. This suggests that the Cu^2+^ and Pb^2+^ adsorption process is chemisorbed by monomolecular layer MgO@ZnO@BC, and the adsorption process may also consist of valence forces between the metal ions and MgO@ZnO@BC through shared electrons.

### 2.4. Mechanism Analysis

The MgO@ZnO@BC adsorption mechanism is shown in Figure 8 and consists of three main components.

(1)Ion exchange: Mg^2+^ and Zn^2+^ are released from the MgO and ZnO groups loaded on the surface of biomass carbon to exchange ions with Pb^2+^ and Cu^2+^ in solution, resulting in PbO, CuO and Pb(OH)_2_, Cu(OH)_2_, etc., which are adsorbed on the surface of biomass carbon and participate in further precipitation reactions. This process is also one of the reactions for precipitation generation. Therefore, the role of ion exchange in the adsorption process needs to be further verified.(2)Precipitation: Pb(OH)_2_, Cu(OH)_2_ and other components have a certain solubility in aqueous solution, and then react with CO_2_ to produce further precipitation of alkaline lead carbonate based on Pb_3_(CO_3_)_2_(OH)_2_ and Cu_2_(OH)_2_CO_3_ and alkaline copper carbonate, greatly increasing the removal rate of Pb and Cu. At the same time, the pH of the system increases due to the action of MgO/Mg(OH)_2_ and CuO/Cu(OH)_2_, which reached favorable conditions for precipitation formation, especially on the surface of the adsorbent, thus allowing formation of lead and copper-containing precipitates in large quantities and adherence to the surface of the biochar, as well as binding to the adsorbent, thus allowing their removal from the solution through the filter membrane. Precipitation plays a major role in the adsorption of Pb(II) and Cu(II) by MgO/Mg(OH)_2_ and CuO/Cu(OH)_2_-loaded materials.(3)The role of surface functional groups: C-containing functional groups such as C-O/C-N, C=O, and O=C-O, as well as some groups which share electrons with Pb(II) and Cu(II) during adsorption, combine with Pb(II) and Cu(II) to achieve removal. The addition of functional groups containing metal oxides by loading metal oxide also plays a role in the adsorption process.

In addition, there are a large number of carboxyl groups in the MgO@ZnO@BC composites that chelate metal ions as a driving effect for adsorption. The electrostatic interaction between the surface of the composite and the metal ions also plays a role in adsorption. Inspired by the adsorption–desorption of HCl, it is perfectly reasonable to synthesize an adsorbent that can generate H^+^ [61]. Temperature can also change the conversion of deprotonation (-COOH) groups in MgO@ZnO@BC composites. Temperature can modify desorption, and at 60 °C, the (-COOH) group can dissociate, and a large amount of H^+^ is produced in the solution. This material without chemical production is promising for environmental applications.

### 2.5. Chemical Simulation Calculations

Traditional experimental methods make it difficult to determine the strength and mechanism of interfacial molecular interactions. With the development of computers and algorithms, molecular simulations have shown great potential in dealing with interfacial interactions and revealing microscopic mechanisms. Compared with other computational simulation software such as NAMD, GROMACS, and LAMMPS, Materials Studio provides a friendly simulation environment for an in-depth study of the properties and processes of various nanoclusters, small molecules, crystalline and amorphous states, and polymeric materials. In this simulation, we constructed MgO@ZnO@BC by Materials Studio 2019, software and calculated the adsorption conformation, energy band structure, and projected density of states of the adsorbent through theoretical calculations to better understand the interaction mechanism of the adsorbent with heavy metal molecules [62]. First, a three-dimensional model was constructed. As shown in Figure 9, the structure of MgO@ZnO@BC was obtained from the random distribution of groups on the surface. All the super-monomers of MgO@ZnO@BC have exact dimensions and contain Mg atoms, O atoms, Zn atoms and C atoms. To facilitate the analysis of the interaction mechanism, defects were not considered in the model of this paper.

Second, a laminar model of the composite interaction was also constructed. In constructing the laminate structure, the initial configuration of the simulation was obtained by geometrical optimization of the model due to periodic boundary conditions.

The energy band structure of the entire simulated process is shown in Figure 10a, and the total and divisional densities of the states are shown in Figure 10b,c. The total density of states (DOSs) and partial density of states (DOSs) as well as the energy band structure of MgO@ZnO@BC are plotted. In Figure 10a, we calculate the energy band structure for the equilibrium parameters of MgO@ZnO@BC. Figure 10a (Z, A, M, G, Z, R, X, and G) represents the different energy levels of the spins obtained in the KS-DFT equation solved in the calculation code. The solid red line shows the first BZ along the symmetry point. It also indicates its Fermi energy level of 0 (E_F_ = 0.0 eV). From the figure, we can see that the GGA potentials of the composites all give rise to semi-metallic states and there is little difference in the structure of the spin states when it comes to E_F_. The energy band follows the spin-up direction through the E_F_, and all the spin bands pass through the E_F_ to produce an energy gap. As the number of electrons in the energy band increases, the space for spin splitting in the spin band is at the upper plate. The additional electrons create a repulsive effect that moves the electrons in the crystal away from the E_F_ and open the gap in the spin–spin band direction. The energy band structure spin-up and the spin–spin optical band are less affected by GGA and are mainly influenced by the rare earth dot position. Both the valence electrons of the surface active oxygen atoms and their interacting atoms show a significant overlap near the Fermi energy level. It is further confirmed that the adsorption of heavy metals by the composites is dominated by chemisorption.

The density of states of an electron is a function that counts the number of different states through which an electron passes by units of volume and per unit of energy at a given energy level. Therefore, to explain the main electronic properties of MgO@ZnO@BC at equilibrium, the total density of states and the partial fractional density of states in the spin-up direction are calculated using the model as shown in Figure 10b,c. From the TDOSs in Figure 10b, it is clear that there are a large number of spin-up TDOS near (E_F_ = 0.0 eV), while the spin-dn TDOSs are zero, indicating the HM nature of the composite. The potential of the highest peak of the spin of the TDOS of the composite reaches a maximum at E_F_ (20 states/eV). Mg, Zn, and C contribute the largest amount of TDOS in the composite, with a small fraction obtained by sharing O atoms. The same can also be seen in the valence band with a part of a small number of peaks further away from the E_F_.

In addition, we also calculated the fractional density of PDOS states for the main electronic hybridization of the composites at different energy levels. Figure 10c shows the fractional density of the states of the different elements in the orbitals. All calculations were performed around the energy band above the mainly localized electronic hybridization of the composites. The valence states show some occupied segregated states with energies mainly concentrated between −18.0 eV and −75.0 eV. It can be seen that the energy of each orbital is the same and consistent with the total fractional density of the energy of the states. The corresponding atoms interact with the surface O atoms while maintaining the balance of surface interactions with the atoms. The better overall alignment of Mg with the O surface and Zn and C with the Mg surface during the motion suggests the reason for the larger adsorption energy of MgO@ZnO@BC.

### 2.6. New Perspectives on Materials

The use of adsorbents should be considered not only in terms of secondary pollution but also in terms of renewability [63]. To better express the potential of MgO@ZnO@BC adsorbents in wastewater treatment, we propose the DQSE (Demand Problem Strategy Expansion) framework as shown in Figure 11, which allows us better visualization of the problems, needs, and national policies, solutions and applications of MgO@ZnO@BC adsorbents. First, the reuse of biomass waste magnesite and wastewater treatment is the driving factor for the MgO@ZnO@BC adsorbent material. Although the MgO@ZnO@BC adsorption material has a good adsorption effect, there are still some problems, such as low adsorption rate, easy-to-reach saturation adsorption, and narrow application range. Therefore, we should make corresponding strategies, such as MgO@ZnO@BC adsorbent materials for process control to solve the problem of easy-to-reach saturation adsorption or to improve the adsorption effect by material modification. Finally, we should promote application in combination with exhaust gas treatment and soil remediation, etc. [64]. Overall, the MgO@ZnO@BC adsorbent material is environmentally friendly, cheap, can reduce the emission of biomass waste as well as waste ore, and can prevent gas and soil pollution. Therefore, it is beneficial to the sustainable development of our country.

## 3. Materials and Methods

The magnesium ore was obtained from the Liaodong area, Liaoning Province. Precursors using raw bananas were purchased from a local supermarket (Hefei, Anhui Province). Zinc acetate, sodium hydroxide, concentrated hydrochloric acid, ammonium carbonate, and anhydrous ethanol were purchased from Sinopharm Co. (Beijing, China). The deionized water was made in the Hefei university laboratory. The stock solution of Cu(II) and Pb(II) was prepared and diluted with deionized water to the desired concentration.

### 3.1. Preparation of MgO@ZnO@BC

First, unlike the traditional grinding process, a special chemical method is used to treat the biomass carbon with HCl impregnation as the pore forming agent [49]. The advantage of the impregnation method is that the good fluidity of the solution allows the HCl full entrance to the original crevices of the banana peel. The preparation steps are as follows: A certain amount of banana peel waste is washed with deionized water, cut into pieces, dried in an oven at 105 °C for 8 h, and the sample is soaked in 100 mL of 6 mol/L of hydrochloric acid for 12 h. The sample is placed in a crucible and calcined in a muffle furnace at 360 °C for 3 h, naturally cooled to room temperature; then, the sample is ground and the fine powder os placed in 100 mL of 6 mol/L of sodium hydroxide solution and stirred for 3 h.

A certain amount of magnesium ore is poured into the muffle furnace at 650 °C calcination for 30 min, cooled to room temperature, and ground into powder using a 200-mesh sieve to obtain magnesium oxide products. The synthesis of ZnO powder by hydrothermal synthesis is conducted as reported in the literature [50]. The prepared magnesium oxide and zinc oxide powders are prepared in a molar ratio to a total volume of 100 mL of aqueous solution and stirred for 10 min. Then, the mixed solution is placed in a microwave synthesizer and heated to 80 °C for 30 min. After the reaction, it is aged at room temperature for 2 h, extracted, filtered, and washed. It is then put into the drying oven at 150 °C for 3 h. The white powder of MgO@ZnO is then obtained. Then, it is put into a muffle furnace to calcine at 650 °C for 1 h to obtain the composite material of MgO@ZnO only.

The banana MgO@ZnO@BC ternary composite was improved in a previous work [27,51]. The mixture was soaked in hydrochloric acid with a molar ratio and dissolved completely, and then stirred in sodium hydroxide solution for 3 h. The mixture was placed in a reaction kettle for 24 h of hydrothermal reaction at 120 °C in an oven. After being cooled to room temperature, the mixture was filtered and washed to neutral. After drying at 105 °C for 6 h, the MgO@ZnO@BC compounds were obtained.

### 3.2. Material Characterization

The phase analysis of the samples was carried out using a SmartLab powder X-ray diffractometer (XRD, working current 10 mA, tube voltage 40 kV, Cu Kα target radiation, λ = 0.15406 nm). IRAffinity-1S Fourier transform infrared spectrometer was used for analysis of functional surface groups. SEM-EDX analyzed the surface morphology and elements of the composites. X-ray photoelectron spectroscopy (XPS) analysis of the surface chemical structure of composite materials was conducted. The thermal stability of the MgO@ZnO@BC adsorbent was analyzed by TGA.

### 3.3. Adsorption Studies

A mixture of Cu^2+^ and Pb^2+^ was prepared to test the adsorption performance of the MgO@ZnO@BC adsorbent. Adsorption studies were carried out under static conditions and 250 mL conical flasks were used for the experiments. Experimental parameters such as the adsorbent dosage, different concentrations of the solution, adsorption time, and different temperatures were discussed to investigate the optimal conditions for adsorption. The supernatant of the solution was taken throughout the adsorption process, and the equilibrium concentrations of Cu^2+^ and Pb^2+^ were measured by principle absorption spectrophotometry.

After adsorption was complete, the equilibrium amounts of adsorption and the adsorption efficiencies of Cu^2+^ and Pb^2+^ were calculated according to Equations (1) and (2).
Q = (C_0_ − C_e_)V/m,(1)
E = (C_0_ − C_e_)/C_0_ × 100%.(2)

C_0_, C_e_, V and m represented in the formula denote the initial and equilibrium concentrations of Cu^2+^ and Pb^2+^, respectively, the solution volumes of Cu^2+^, Pb^2+^ and the amount of the MgO@ZnO@BC added adsorbent.

### 3.4. Statistical Analysis

Adsorption experiments were performed in three replicate cycles. Data on adsorption volume from the adsorption experiments were subjected to one-way ANOVA for statistical analysis and were statistically significant when *p* < 0.05.

## 4. Conclusions

In summary, the first MgO@ZnO@BC ternary composite adsorbent was synthesized for the adsorption of heavy metals in wastewater by hydrothermal synthesis. The results show that the addition of the MgO@ZnO@BC composite has higher adsorption properties for Cu^2+^ and Pb^2+^, with a molar ratio of 5% 0.1 g, and maximum adsorption capacity (50.63 mg/g for Cu^2+^ and 61.46 mg/g for Pb^2+^) at 273 K. The Langmuir isotherm model and the pseudo-secondary kinetic model fit best and can be used to describe the adsorption mechanism. The adsorption mechanism mainly includes ion exchange, functional group complexation, precipitation, etc. It was shown that MgO@ZnO@BC is effective and feasible for the removal of heavy metals from industrial wastewater. MD simulations allow the study of the interaction between different elements, and it was further confirmed that the adsorption is mainly chemisorption.

## Figures and Tables

**Figure 1 molecules-28-06982-f001:**
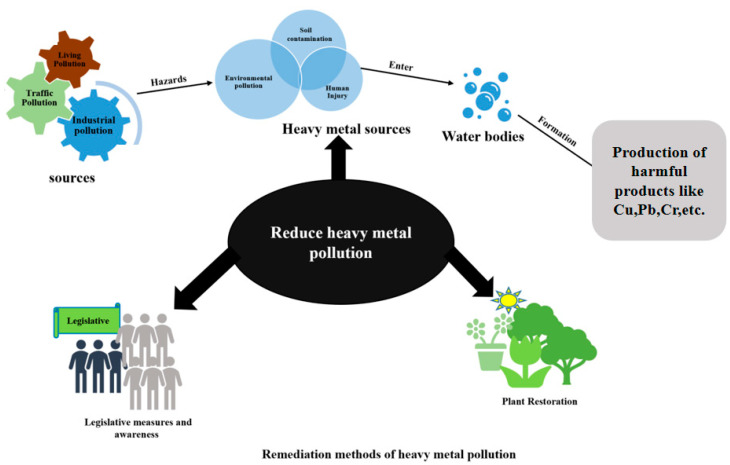
Remediation methods for heavy metal pollution.

**Figure 2 molecules-28-06982-f002:**
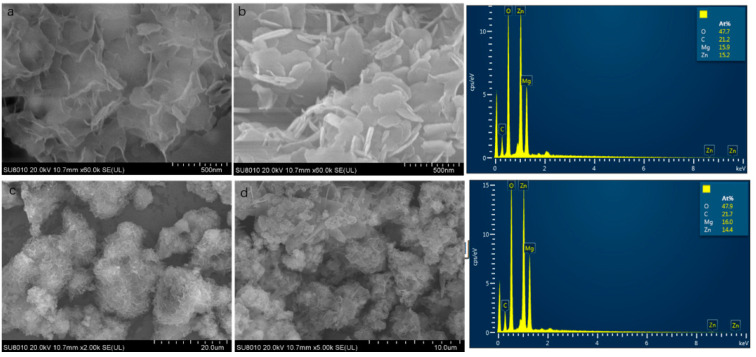
MgO@ZnO@BC adsorbent SEM images, (**a**) amplify in 60,000 times, (**b**) amplify different parts by 60,000 times, (**c**) amplify in 2000 times, (**d**) amplify in 5000 times, and EDX spectrum.

**Figure 3 molecules-28-06982-f003:**
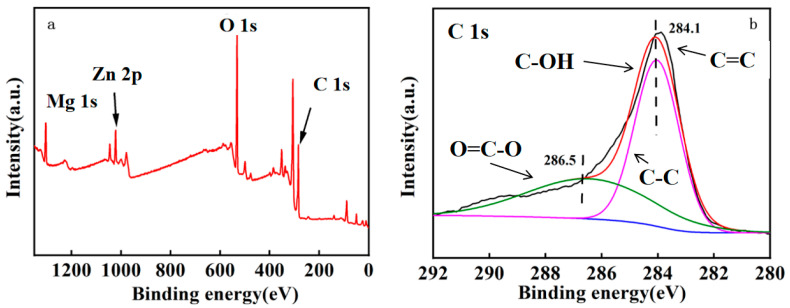
XPS scan spectra (**a**,**b**) of MgO@ZnO@BC ternary composite adsorbent.

**Figure 4 molecules-28-06982-f004:**
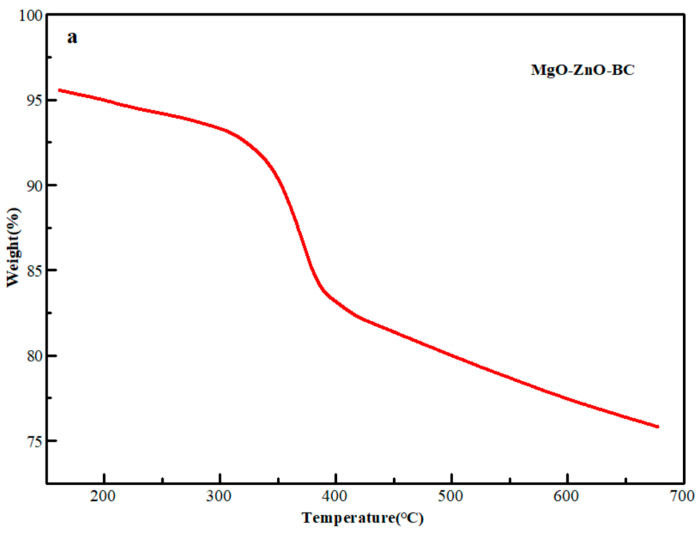
Plots of MgO@ZnO@BC adsorbent. (**a**)TG images, (**b**) XRD spectrum, (**c**) FT-IR spectrum.

**Figure 5 molecules-28-06982-f005:**
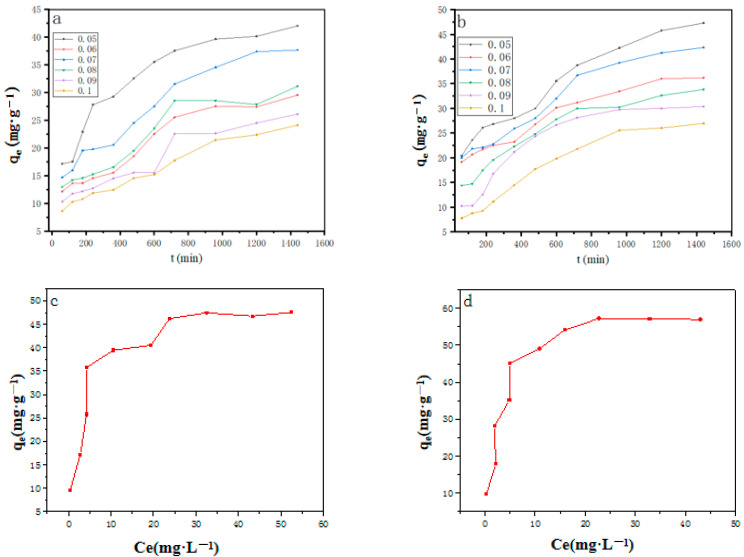
(**a**,**b**) Adsorption effect of different adsorbent times on Cu^2+^ and Pb^2+^; (**c**,**d**) Adsorption effect of adsorbents on Cu^2+^ and Pb^2+^ at different initial concentrations; (**e**,**f**) Adsorption effect of adsorbents on (**e**) Cu^2+^ and (**f**) Pb^2+^ at different adsorbent masses; (**g**) Effect of temperature on the adsorption effect.

**Figure 6 molecules-28-06982-f006:**
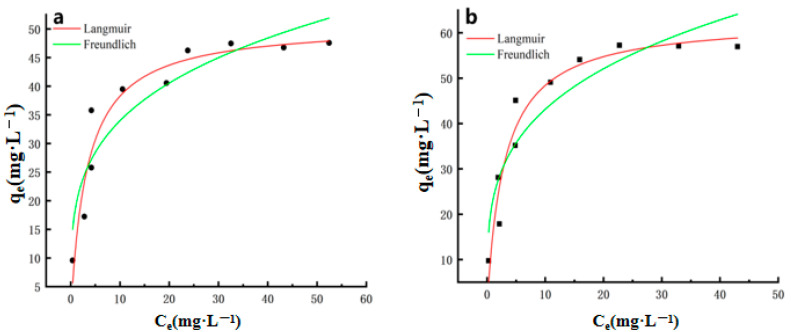
Langmuir and Freundlich isothermal adsorption model for (**a**) Cu^2+^ adsorbent, (**b**) Pb^2+^ adsorbent.

**Figure 7 molecules-28-06982-f007:**
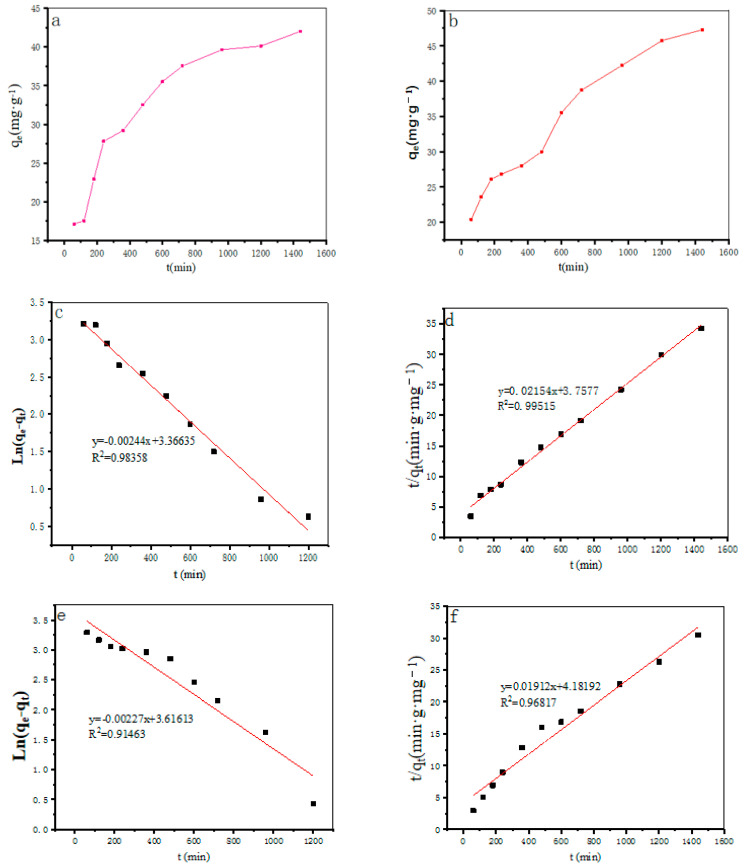
(**a**,**b**) Adsorption effect of 5% adsorbent on Cu^2+^ and Pb^2+^; (**c**) primary kinetic simulation of adsorbent on Cu^2+^; (**d**) secondary kinetic simulation of adsorbent on Cu^2+^; (**e**) primary kinetic simulation of adsorbent on Pb^2+^; (**f**) secondary kinetic simulation of adsorbent on Pb^2+^.

**Figure 8 molecules-28-06982-f008:**
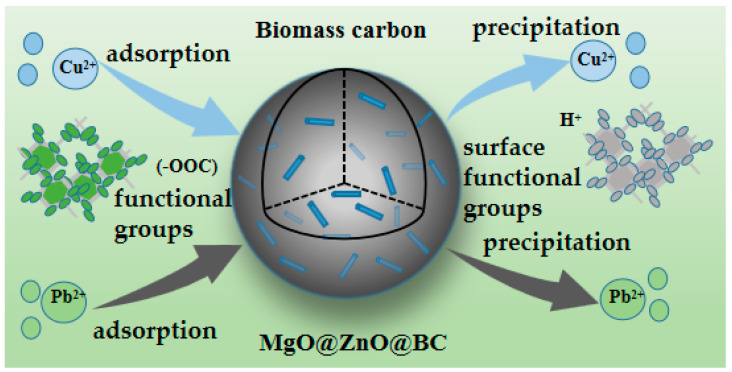
Schematic diagram of the adsorption mechanism of MgO@ZnO@BC.

**Figure 9 molecules-28-06982-f009:**
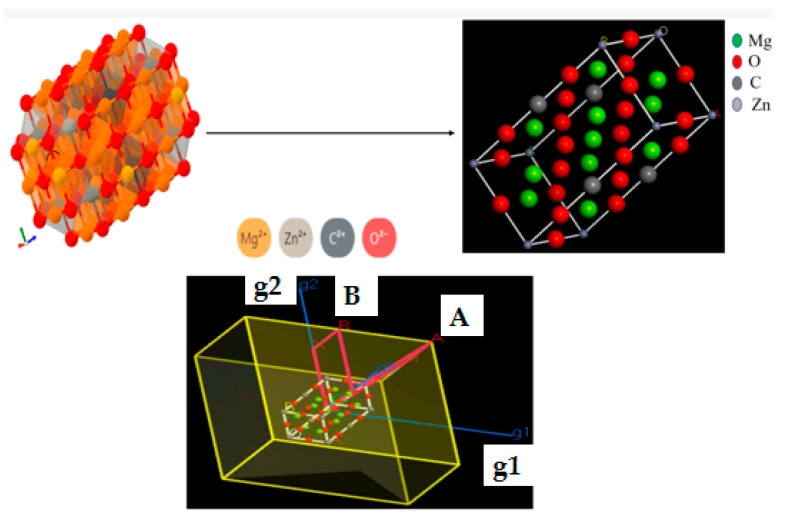
MgO@ZnO@BC adsorption model.

**Figure 10 molecules-28-06982-f010:**
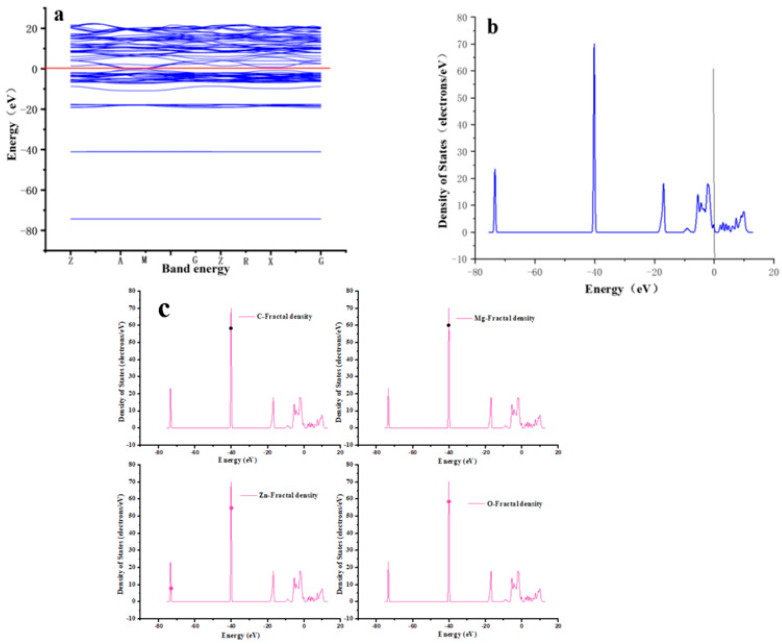
(**a**) is the energy band of the MgO@ZnO@BC adsorbent, (**b**) is the total density, (**c**) is the fractional density of individual elements.

**Figure 11 molecules-28-06982-f011:**
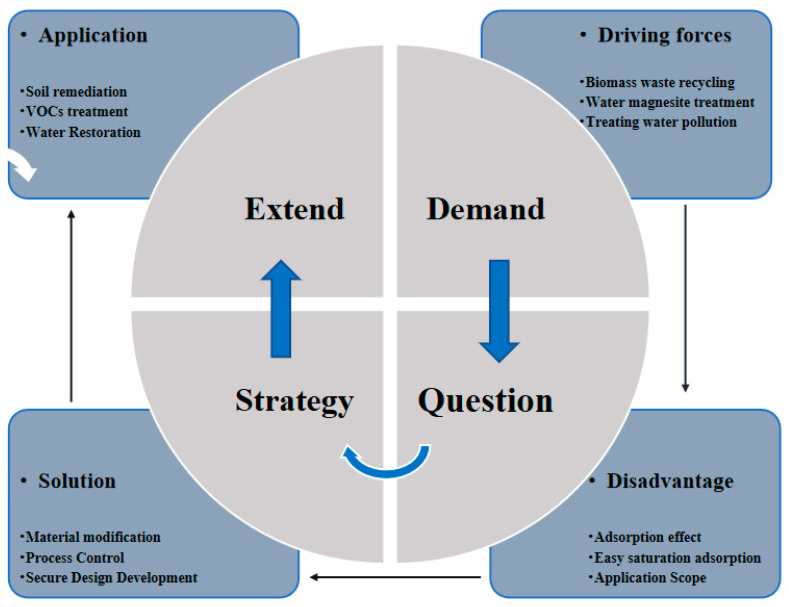
DQSE framework (Demand Problem Strategy Extension) for long-term tailing applications for waste applications.

**Table 1 molecules-28-06982-t001:** Relevant values of the data fitting model.

Adsorbent	Langmuir			Freundlich		
	q_max_ (mg·g^−1^)	K_L_ (L·mg^−1^)	R^2^	K_F_ (mg^1−*n*^·L^n^·g^−1^)	*n*	R^2^
Cu^2+^	50.89628	0.30251	0.92517	18.85133	3.91068	0.8653
Pb^2+^	63.00477	0.33164	0.94099	23,0983	3.68433	0.8688

## Data Availability

Not applicable.

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
