# Peer review of "Preparation of Biomass Carbon Composites MgO@ZnO@BC and Its Adsorption and Removal of Cu(II) and Pb(II) in Wastewater"

_molecules, 2023, doi:10.3390/molecules28196982_

Round 1

Reviewer 1 Report

Comments and Suggestions for Authors

In this work, a ternary composite MgO/ZnO/BC was synthesized and characterized for the adsorption of heavy metal ions, Cu2+ and Pb2+, from wastewater. Although there are some interesting findings, the manuscript is still very immature and has obvious flaws. Thus, it can’t be accepted for publication in its present form.

1.      In the introduction, the author does not explain clearly why it is necessary to use ternary adsorbent to complete the adsorption task of copper and lead ions. In other words, what advantages do ternary adsorbents have over binary or single BC adsorbents for Cu2+ and Pb2+ adsorption?

2.      The most intolerable shortcoming in the present manuscript is that no specific absorption spectrophotometry method of measuring Cu2+ and Pb2+, respectively, was given in the 2.3 section. Therefore, I seriously doubt the authenticity of these adsorption data.

3.      The sentences are poorly organized and difficult to read.

4.      Fig.1, Fig.9 and Fig.11 have no any relevance to this topic.

Comments on the Quality of English Language

English very difficult to understand/incomprehensible.

Author Response

Dear Editors and reviewers,

I am very grateful to your comments for the manuscript. Your suggestions can make this article more rigorous and complete. According with your advice, we amended the relevant part in manuscript. Some of your questions were answered below.

Reviewer 2 Report

Comments and Suggestions for Authors

The novelty of the work is poor, wrt synthesis and the end application. There are several reports available on the heavy metal sorption using sorbents which can be easily synthesized, recovered with high sorption capacity. Most of the text is inspired from Arabian Journal of Chemistry Volume 15, Issue 9, September 2022, 104059. The current work doesn't show any importance with respect to synthesis, characterization and sorption capacity. Too many figures are included and key figures are missing wrt characterization and sorption. Control expts are missing which are imp wrt synthesized material. 

No necessity of Fig 1. The synthesis looks complicated. Discussion related to SEM is totally baseless. Why only C1s scan is provided in XPS section? No comparisons with state of art heavy metal sorption. I recommend the rejection of the manuscript in the current form.

Comments on the Quality of English Language

Throughout the manuscript it needs corrections. Too many typos are present. Figure captions are bad.

Author Response

Response: Thank you for the suggestions.

Dear Editors and reviewers,

We are very grateful to your comments for the manuscript. Your suggestions can make this article more rigorous and complete. According with your advice, we amended the relevant part in manuscript. Some of your questions were answered below.

Although MgO@ZnO@BC adsorption material has a good adsorption effect, there are still some problems. Low adsorption rate,easy to reach saturation adsorption,and narrow application range. So,we should make corresponding strategies, such as MgO@ZnO@BC adsorbent materials for process control to solve the problem of easy-to-reach saturation adsorption, or to improve the adsorption effect by material modification.Finally,we promote the application in combination with exhaust gas treatment and soil remediation,etc(Xu et al. 2022).

Xu Y J,Xia H Y,et al.Adsorption of cadmium(II) in wastewater by magnesium oxide modified biochar.Arabian Journal of Chemistry,2022,15(9):104059

We have made correction according to the Reviewer’s comments.

Reviewer 3 Report

Comments and Suggestions for Authors

The adsorption performance of MgO@ZnO@BC biomass carbon composites on Cu2+ and Pb2+

1.       The abstract should be specific, highlighting the main objectives.

2.       The rational use of metal oxide and carbon can be addressed in introduction section with the integrity of following articles: Chemical Physics Letters, Volume 804, October 2022, 139884, Journal of Energy Storage, Volume 60, April 2023, 106713

3.       The C 1s fitting is not correct. Mg 1s, Zn 2p and O 1s also should be explained. In XPS, C1s, should be C 1s.

4.       JCPDS card should be given to the XRD patterns. In Y axis-intensity should be given.

5.       Nonlinear fitting should be done for the adsorption kinetics and adsorption isotherms with the help of different models: For reference, Chemical Engineering Journal, Volume 417, 1 August 2021, 129312, Separation and Purification Technology, Volume 287, 15 April 2022, 120463

6.       The adsorption mechanism can be explained based on the post XPS and FTIR.  

Comments on the Quality of English Language

Minor grammar check and word spacing. In XPS, C1s, should be C 1s.

Author Response

Response: Thank you for the suggestions.

Dear Editors and reviewers,

We are very grateful to your comments for the manuscript. Your suggestions can make this article more rigorous and complete. According with your advice, we amended the relevant part in manuscript.We have made correction according to the Reviewer’s comments. Some of your questions were answered below.

Round 2

Reviewer 1 Report

Comments and Suggestions for Authors

After closely inspecting the revised MS and the detailed responses to the reviewers, I have noticed that the most of problems concerted by the reviewers have been addressed. Thus, I recommended this article publication.

Comments on the Quality of English Language

Minor editing of English language required.

Author Response

Thank you for the suggestions.

Dear Editors and reviewers,

We are very grateful to your comments for the manuscript. Your suggestions can make this article more rigorous and complete. According with your advice, the English version of the paper has been revised and edited.Revised portion are marked in red in the paper.

Reviewer 2 Report

Comments and Suggestions for Authors

The manuscript needs considerable changes with respect to image, discussion and justification. The points are given below.

Figure caption 2 needs to be rewritten mentioning all the images and discussion.

Why only C1s scan is provided in XPS section? No comparisons with state of art heavy metal sorption.

Osher peak??

plus a nuclear level peak of additional Zn What is nuclear level peak??

Fig 5 needs to be redrawn clearly which should suite a manuscript. The present images are placed not well.

Same thing for Fig 7.

Evidences related to Fig 8 needs to be included.

What exactly authors want to highlight from Chemical simulation image? MgO@ZnO@BC adsorption model? The figure captions are poor mostly.

Fig 10 needs to be redrawn carefully.

Comments on the Quality of English Language

Language needs to be improved

Author Response

Thank you for the suggestions.

Dear Editors and reviewers,

We are very grateful to your comments for the manuscript. Your suggestions can make this article more rigorous and complete. According with your advice, we amended the relevant part in manuscript. We have studied comments carefully and have made correction which we hope meet with approval. Revised portion are marked in red in the paper. The main corrections in the paper and the responds to the reviewer’s comments are as flowing:

Reviewer 3 Report

Comments and Suggestions for Authors

Accept

Author Response

(The authors gave the same response as above.)
